



# Satellite data validation: a parametrization of the natural variability of atmospheric mixing ratios

Alexandra Laeng[1], Thomas von Clarmann[1], Quentin Errera[2], Udo Grabowski[1], and Shawn Honomichl[3]

[1]Karlsruhe Institute of Technology, Institute of Meteorology and Climate Research, Karlsruhe, Germany
[2]Royal Belgian Institute of Space Aeronomy, Brussels, Belgium
[3]UCAR/NCAR/ACOM, Boulder, Colorado, USA

**Correspondence:** A. Laeng (alexandra.laeng@kit.edu)

**Abstract.** High-resolution model data are used to estimate statistically typical variabilities of mixing ratios of trace species as a function of spatial and temporal distance. These estimates can be used to explain that part of the differences between observations made with different observing systems that are due to less than perfect collocation of the measurements. The variability function is approximated by a two-parameter regression function, and look-up tables on the natural varaibility
as a function of distance separation and time separation are provided. In addition, a reparametrization of the variabilities values as function of latitudinal gradients is proposed, and season-independence of linear approximation of such function is demonstrated.

## 1 Introduction

This paper tackles a problem which typically arises when remotely sensed data from different instruments are compared within
the framework of validation studies. In quantitative validation, the common approach is to calculate differences of pairs of measurements of the same airmass by the two instruments under comparison. With the aid of $\chi^2$-statistics it is tested if the observed differences can be explained by the estimated error of the differences (Rodgers and Connor, 2003). The estimated error of the differences includes measurement noise and parameter as well as model errors, as far as they are uncorrelated between the instruments (von Clarmann, 2006). Also different impact of prior information on the result has to be considered.
But still, often, the differences will be too large to be explained by the combined error budget of the measurements under comparison. The reason is that the instruments typically do not sound exactly the same airmass. Spatial and temporal mismatch of the measurements along with natural variability of the state variable measured contributes to the observed differences. This source of differences is quantified in validation papers only in a few exceptional cases (see, e.g., Sheese et al. (2021) for an example where models were used to quantify the related effect of ozone variability). Instead, natural variability is often
used as a universal excuse to defend measurements of which the validation studies suggest that the related retrieval errors are underestimated.

In this study, we present a user-friendly tool to provide quantitative estimates of that component of the differences of observations which can be attributed to the spatial and temporal mismatch and natural variability. The underlying method is based on high-resolved model fields of temperature and mixing ratios of trace species as described in Sect. 2. From these fields





the typical variabilities are evaluated as a function of spatial and temporal mismatch (Sect. 3). In order to avoid unnecessary
large data traffic and to reduce as much as possible the impact of model imperfections on the calculated fields, parametriza-
tions of these dependencies, for different trace gases, altitudes, and latitude bins, are developed, by prescribing to the natural
vairability function a particular shape, arising from general theory of random functions with stationnary increments (Sect. 4)
and confirmed by calculations out of model data. A re-parametrization is developed to improve the validity of the inductive
generalization towards other gases and seasons (Sect. 5). A recepee how to use this reparametrization is descibed in Sect. 6.
The adequacy of our suggested method is critically discussed in Sect. 7 and final recommendations are given in Sect. 8.

## 2  Model data

Model fields used in this study came frop two models.

### 2.1  BASCOE

Main set of fields has been produced by the Belgian Assimilation System for Chemical ObsErvations (BASCOE, Errera et al.
(2008)). While usually used in the context of stratospheric chemical data assimilation (e.g.Errera et al. (2019) ), the Chemistry
Transport Model (CTM) of the system has been also used to study of the evolution of the stratospheric composition (Chabrillat
et al., 2018; Prignon et al., 2019; Minganti et al., 2020). Here, BASCOE was run for the period 25 Sept-1 Oct 2008 where wind
and temperature were taken from the ERA-Interim reanalysis (Dee et al., 2011). The model was run on an $1° \times 1°$ horizontal
grid, the native 60 vertical levels of ERA-Interim (from the surface to 0.1 hPa) with a time step of 30 minutes. Hourly global
fields of 28 relevant trace gases[1] were used for this study.

### 2.2  WACCM6

Auxilliary set of model fields used came from Whole Atmosphere Community Climate Model 6 (WACCM6) which is the
atmospheric component of the Community Earth System Model, Version 2 (CESM2) (Danabasoglu et al., 2019; Emmons
et al., 2019; Gettelman et al., 2019; Tilmes et al., 2019) run in UCAR/NCAR/ACOM. The model has a horizontal resolution
of $0.9° \times 1.25°$ degrees with 88 vertical hybrid sigma-pressure levels, and is run using specified dynamics, with nudging of
temperature, and U/V winds from the NASA Goddard Earth Observing System, Version 5 (GEOS-5) forecast model. They
contain the fields of 3 species ($O_3$, $H_2O$, $NO$) for 4 weeks in year 2020, one in each season. The data were regridded on the
same fixed height grid with 1-km step, using the fields of geopotential heights and temperature, provided with the data.

---

[1]$BrNO$, $BrO$, $CCl_4$, $CFC_{11}$, $CFC_{12}$, $CFC_{113}$, $CH_3Cl$, $CH_4$, $ClO$, $ClONO_2$, $CO$, $CO_2$, $H_2O$, $HBr$, $HCl$, $HF$, $HNO_3$, $HNO_4$, $HO_2$, $HOBr$, $HOCl$, $N_2O$,
$N_2O_5$, $NO$, $NO_2$, $NO_3$, $O_3$, $OH$, and temperature




## 3  Variabilities

### 3.1  Structure functions

Let $X$ be a random variable defined by the amount of the target trace gas in a given infinitely small air parcel, centered around a point in the atmosphere at a given moment of time, reported in volume mixing ratio (vmr). The amount of the trace gas in any point of the atmosphere at a fixed time (or in any moment of time in a fixed point of the atmosphere) can be viewed as a

state of a one-dimensional random process $X(t)$, where $t$ parametrizes distance from an initial point (or time elapsed from an initial moment). This random process is not stationnary, because its statistical characteristics can change with $t$. The increments $X(t+\tau)-X(t)$ of the process $X(t)$ represent the change of the amount of the trace gas over distance (or over time). In a given sufficiently narrow latitude band, at a fixed altitude, and in a given season, the distribution of the differences $X(t+\tau)-X(t)$ does not depend on $t$, which means that $X(t)$ is the process with stationnary increments. The basic characteristics of real-

valued random process with stationnary increments are the mean value of the increment $E[X(t+\tau)-X(t)]$ and the correlation function of the increment

$$D(\tau) = E|X(t+\tau) - X(t)|^2 \tag{1}$$

also called *the structure function* of the process $X(t)$ ((Yaglom, 1986), ch.23). In our case, as $E[X(t+\tau)-X(t)] = 0$,

$$D(\tau) = E|X(t+\tau) - X(t)|^2 = \sigma^2(|X(t+\tau) - X(t)|) + (E|X(t+\tau) - X(t)|)^2 = \sigma^2(|X(t+\tau) - X(t)|). \tag{2}$$

The natural variability of a trace gas is the square root of the structure function of the process $X(t)$ :

$$\sqrt{D(\tau)} = \sigma(|X(t+\tau) - X(t)|). \tag{3}$$

This provides the formal link between the intuitive definition of the natural variability as the variability of differences and the mathematical machinery of random processes with stationnary increments which is widely used in stydying the processes at smaller spatio-temporal scales, for exemple, in the theory of atmospheric turbulence. This link will allow us to draw conlusions

about the nature of the process $X(t)$ basing on the shape of statistics obtained and will justify the choice of the form of the regression function for natural variability of trace gases. Next section explains how the estimation of $\sqrt{D}$ is done out of model fields.

### 3.2  Estimation of variabilities

In order to obtain a statistic of the variability of differences, in a first step the model fields were transformed from their native

hybrid sigma-pressure vertical grid to a fixed 1-km step geometrical height grid. For this, the geopotential height for each model knot was restored and transformed into geometrical height using the temperature values from the model, which allowed the interpolation of the profiles on a fixed altitude grid. In the second step, the model fields were smoothed according to the horizontal resolution of the instruments whose precision is to be validated. We have chosen as an example the Michelson Interferometer for Passive Atmospheric Sounding (MIPAS, Fischer et al. (2008)). Its cross track resolution corresponds roughly





to an East-West resolution and is defined by the width of the instantaneous field of view at the tangent point, which is 30 km.
The along-track smearing, which corresponds roughly the the North-South horizontal resolution is on average roughly 200 km
(von Clarmann et al., 2009). This smoothing operation does not influence the shape of the obtained curves, but does reduce
the obtained variabilities values by around 0.05%. No vertical smoothing is applied because vertical smoothing typically is
considered in validation in explicit manner via the averaging kernels. These smoothed fields are the basis for the statistics of
horizontal and temporal variability of the atmospheric state.

### 3.2.1 Horizontal Variability

We take the model data within a fixed 10-degree latitude bin and at a fixed height; as each of five model datasets used (one from
BASCOE and four from WACCM6) covers only one week, there is no need to fix a season. For all possible pairs of points in
the obtained subset, the normalized differences of the volume mixing ratio (vmr) of the target trace species within a predefined
radius of 1500 km are calculated:

$$\frac{VMR(location_1, t) - VMR(location_2, t)}{VMR_{mean}}, \tag{4}$$

with $VMR_{mean}$ being the mean VMR values of the target trace gas in the chosen latitude band at the chosen height. The
constant time index $t$ indicates that only differences are considered where the subtractor and the subtrahend refer to the same
time. These differences are binned according to their horizontal separation distance. The following bins were used: 0 to 100 km,
to 200 km, etc between the two points. We calculate the standard deviation of the sample of these normalized differences,
it provides us with an estimator of natural variability of the target trace gase as function of distance separation. The obtained
natural variability of ozone at 35 km altitudes as a function of distance is shown on the left panel of Figure 1. The fast
growth of the variability for separation distances over 1000 km at high northern latitudes reflects that in many pairs of the
corresponding sample, one point lies inside the polar vortex, another lying outside. Note also that the calculated variability
values for the distances 0-100 km is zero for the tropical latitudes (yellow curves in Figure 1), and present a peak at subtropical
latitudes (clear orange and clear green curves in Figure 1). This is due to models' resolutions: the samples for 100 km distance
separation are empty or very small at low latitudes. Therefore the point 100 km will not be taken into account in the calculation
of regression coefficients at these latitudes.

### 3.2.2 Temporal Variability

Similar as above, for all possible pairs of data points of the entire data set the differences of the volume mixing ratio of the
target trace gas within a predefined time period of 72 hours are calculated:

$$\frac{VMR(location, t_1) - VMR(location, t_2)}{VMR_{mean}}. \tag{5}$$

The constant location index $location$ indicates that only differences are considered where the subtractor and the subtrahend
refer to the same location. These differences are sorted according to their time lag. Similar to the horizontal variability, for
each time lag, the differences are normalized by the mean vmr within given latitude band at given altitude, then the standard





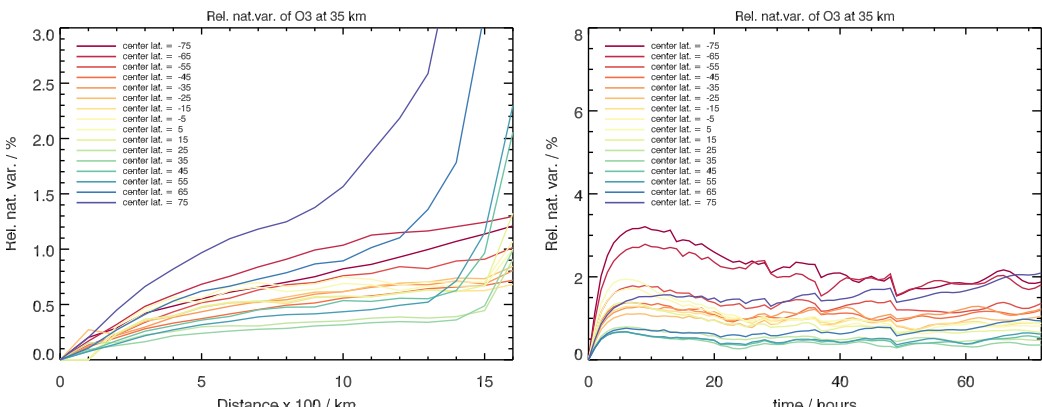

**Figure 1.** Left panel : natural variability of $O_3$ at 35 km altitude as function of horizontal distance. Right panel : natural variability of $O_3$ at 35 km altitude as function of time separation.

deviation of the sample of normalized differences is calculated. This quantity is the estimator of natural variability of the target species as function of time separation; its values for ozone at 35 km altitude are shown on the right panel of Figure 1. As for most of satellite validation exercise, time separation within collocation criteria stays within 5 hours, we made the choice to restrain our analysis to the time separation lag to maximum of 5 hours.

### 3.3 Combination of Horizontal and Temporal Variability

Despite the fact that advection can admittedly cause the correlations between horizontal and temporal components of the variability, at scales considered here, we assume that horizontal and temporal variation of the atmospheric state are uncorrelated. In our analysis we offer independent parametrizations for each of them, which are recommended to be combined by their quadratic sum; we also provide a software performing this summation for a reparametrisation of these quantities on latitudinal

gradients. Tests using a statistic of combined horizontal and temporal differences of the type

$$VMR(location_1, t_1) - VMR(location_2, t_2) \tag{6}$$

have shown that at scales considered here, the error due to the neglect of correlations is below 0.1% and thus not usually worth an additional effort. We are not considering the vertical variability in the present work, the fileds are calculated for each altitude level independently.

## 4 Parametrization

### 4.1 Motivation

The goal of the present work is to provide the community with information on the natural variability of mixing ratios of trace gases as a function of distance and time separation. This information is meant to be used in the context of validation studies.





Instead of providing the entire variability data set, we consider the use of a simple and easy-to-use parametrization as more

adequate. The reasons are these. First, the use of parametrizations avoids a considerable amount of data traffic. Second, the fine structure of the fields reflects the actual conditions of the days actually covered by the model run rather than the general behaviour of the atmosphere. And third, our parametrization by continuous regression functions allow easy interpolation.

## 4.2 The regression function

In view of the shape of the curves produced out of model's data (section 3.2.2 and 3.2.1), the natural variability function

can be paramerized in the form $D(\tau) = A\tau^\gamma$ with $A > 0$ and $0 < \gamma < 1$. An interesting side conclusion that can be made from the obtained shape of the structure function of $X(t)$ is that the process of atmospheric variability (horizontal as well as temporal) of mixing ratios is self-similar: the form of its structure function $D(\tau) = A\tau^\gamma$ is invariant under a group of similarity transformations $t \to ht$, $X \to a(h)X$ (Kolmogorov, 1940; Yaglom, 1986); in other words, no characteristic scale can be associated with their structure function. Note, that atmospheric variability as function of distance and time separation

could be approximately represented by a two-dimensional random process; this is however out of scope of this paper: our choice is to treat the distance and time mismatch dependences separately, because this is what a typical validation exercice does.

As pointed out in the Sect. 3.2.1, at high latitudes at the distances over 1000 km the variability grows very rapidely. Also, for low latitudes, the values of variability as function of distance mismatch is meaningless at 100 km : it is calculated on the

samples from too small to empty, because of the model horizontal resolution. The choice of 5 hours upper limit for the time dependent variability is driven by the typical values of the time mismatch occuring in satellite validation studies, and the shape of the obtained curves. We calculate the regression coefficients $A$ and $\gamma$ by minimising the quantity

$$\sum_{i=2}^{10} (y_i - Ax_i{}^\gamma)^2 \tag{7}$$

via Sequential Least SQuares Programming (SLSQP) optimizer (Kraft, 1988) and by giving the value at 100 km as a first

constraint for $A$ and 0.5 as a first constraint for $\gamma$. The obtained regression curves and the initially calculated model curves for some species are shown in Figure 2 for distance separation and Figure 3 for time separation. The initial and smoothed (regressed) natural variability surfaces as function of latitude and time separation are shown in Figure 4 for ozone at 35 km altitude. In a range between 100 and 1000 km the parametrizations fit the data very well.

## 5 Re-parametrization on latitudinal gradients

An obvious deficiency of our approach is that the variability fields are calculated out of only one week of the data. Operational constraints did not allow to generate a better coverage for these many species at the required resolution. The variability, which we have calculated as function of latitude and distance (time) mismatch, is also season-dependent, because different seasons correspond to different inclinations of the Earth axis and intiuitively all should be schifted in the latitudes while the season changes. There is however a way to encounter the problem: if the user is willing to calculate one additional quantity out of

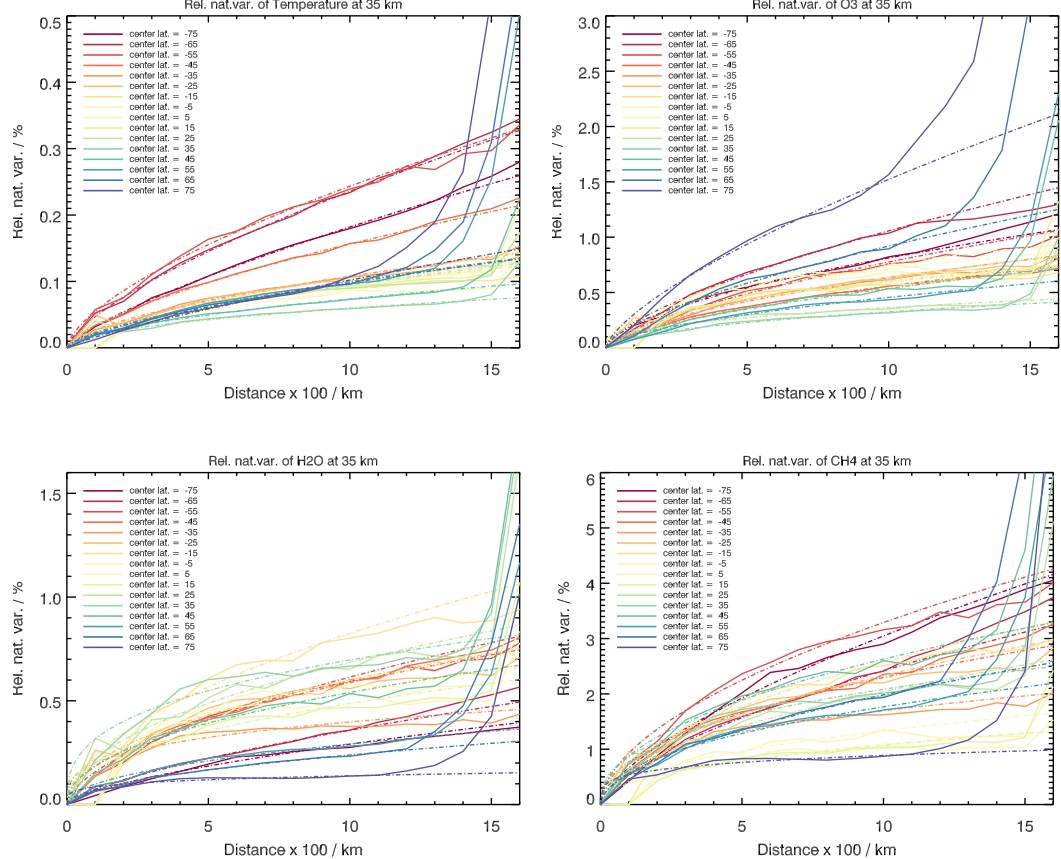

**Figure 2.** Natural variability as function of horizontal distance for temperature, $O_3$, $CH_4$, and $H_2O$ (solid lines) and proposed parametrisation (dashed lines) in different latitude bins.

his data, namely the latitudinal gradients of the gas under validation, then the workaround consists in re-parametrisation of the variability on latitudinal gradients of the species. Latitudinal gradients of a gas are defined as

$$\left| \frac{VMR_{l_1} - VMR_{l_2}}{l_1 - l_2} \right|, \tag{8}$$

where $VMR_{l_1}$ is the mean $VMR$ of the gas in a latitude band, $VMR_{l_2}$ is the mean $VMR$ in the northwise neighboring latitude band, $l_1 - l_2$ is the width of the latitude band, in our case this is 10 degrees. In relative version of latitudinal gradient, this quantity is normalized with respect to the mean $VMR$ in both bands, and is multiplied by 100. For a fixed distance (or time) mismatch $x$, in a first approximation, as the variability is calculated as the square root of the variance of the squares of differences, and the latitudinal gradients are calculated as differences, we expect a linear dependence of the variability from the latitudinal gradients.

To test the theoretical considerations above in practice, especially to see if the linear dependence of the natural variability from the latitudinal gradients is the same in different seasons, the link between the natural variability and the latitudinal gradi-





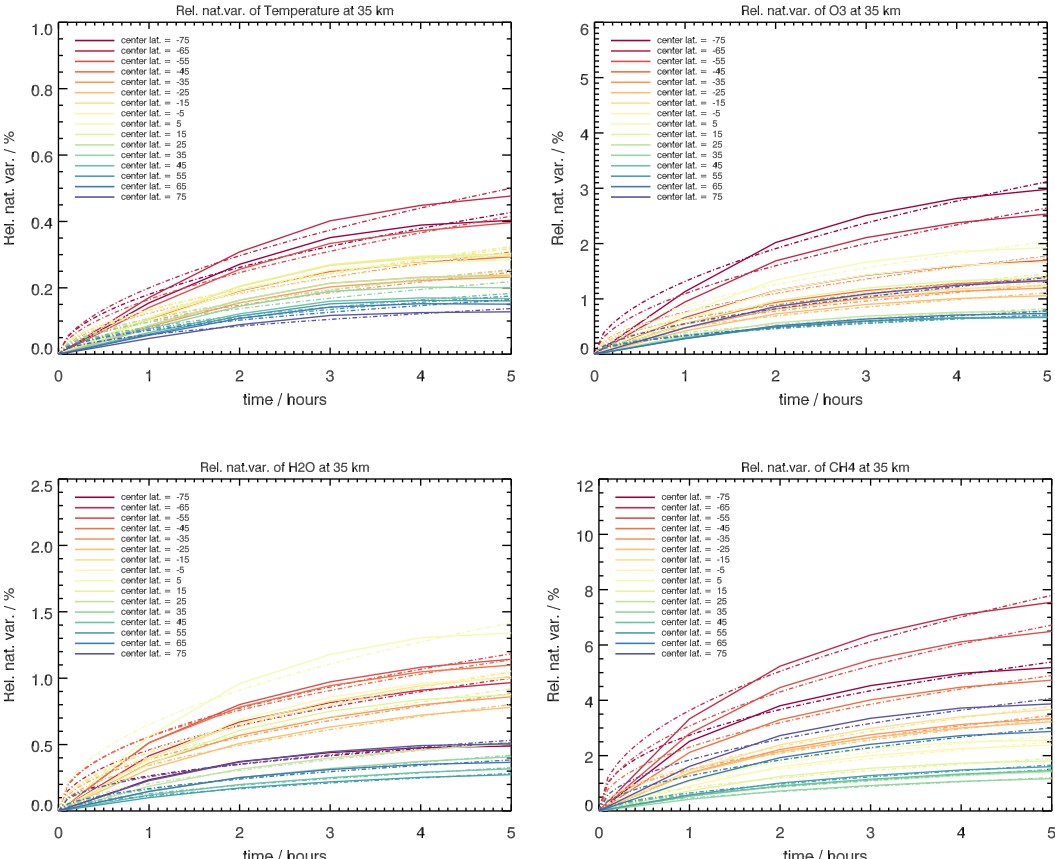

**Figure 3.** Natural variability as function of time separation for temperature, $O_3$, $CH_4$, and $H_2O$ (solid lines) and proposed parametrisation (dashed lines).

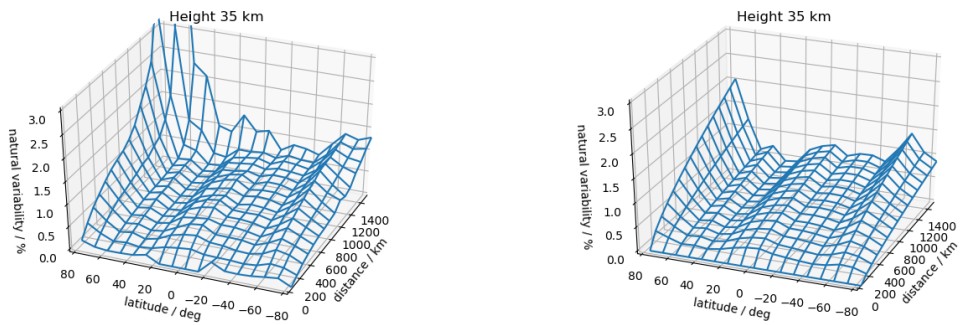

**Figure 4.** Natural variability of $O_3$ at 35 km altitude as function of horizontal distance separation. Left panel : model fields. Right panel : regressed fields.





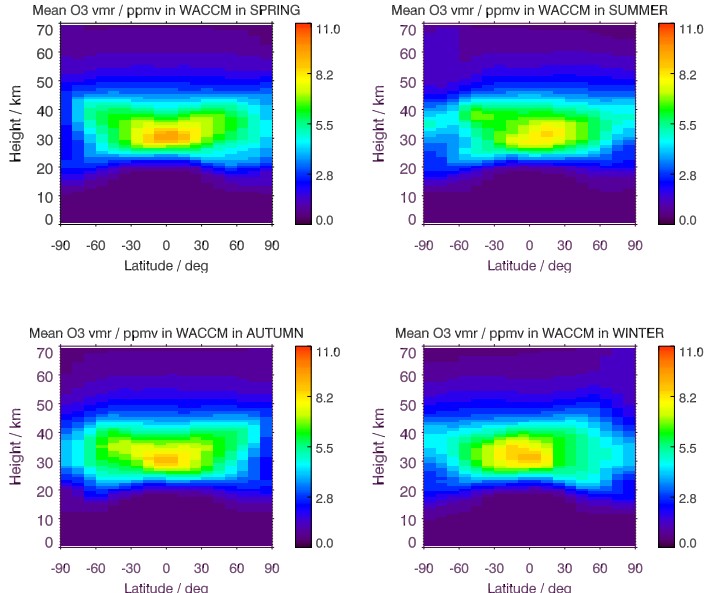

**Figure 5.** Distribution of $O_3$ in one week of each season in WACCM model.

ents was tested on the data of a worse resolved model with less species, but in turn, for which the data from different seasons were available, were available, namely data from the Whole Atmosphere Community Climate Model, Version 6 (WACCM6).

The Figure 5 shows the ozone distribution in each season of WACCM data, after this manipulation of the data : the distributions are as expected, the latitudinal shift in different season is visible, so the data are suitable for our test.

The Figures 6, 8 and 7 show the natural variability as function of latitudinal gradients for a particular distance separation of 400 km in four seasons of WACCM for $H_2O$ at 30 km, $O_3$ at 35 km, and NO at 40 km. Each point in these figures corresponds to one 10-degree latitude bands; for the sake of completeness we include the points from all the latitudes. One can observe that in all cases the regression lines are similar, some deviation in summer and winter come from the points corresponding to high (> 70 degrees) latitudes, which is expectable. This hints toward season-independence of linear approximation of the natural

variability as function of latitudinal gradients. Hence, the natural variability calculated in just one season and reparametrized on latitudinal gradients of the specie, provides all the information needed for the validation exercice in any season. Finally, we would like to remark that the variability fields of well correlated tracers are close, which provides an additional confidence in the method.

## 6 What to do in practice: the software

The variabilites as functions of time and distance mismatch are added quadratically and this provides the final variability value for the collocation criteria choosen. In practice, the users will have to calculate out of their data under validation just one additional quantity, namely the latitudinal gradients of the species under validation. This quantity should be calculated on the

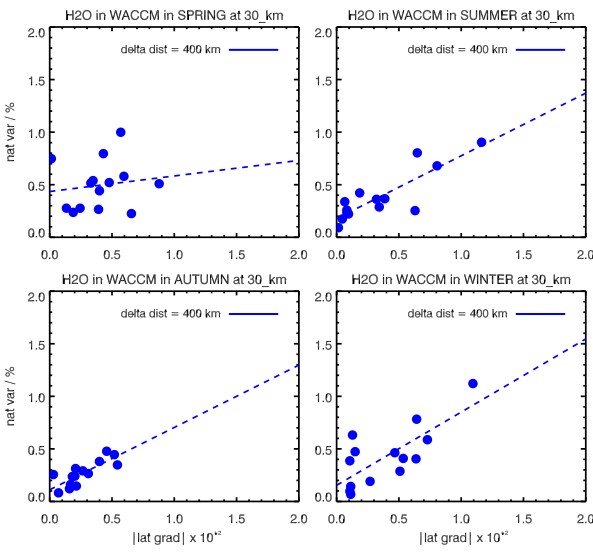

**Figure 6.** Natural variability of $H_2O$ as function of latitudinal gradients for 400 km distance separation at 30 km altitude in four seasons in WACCM model. The horizontal axis corresponds to latitudinal gradients, the dashed line is the linear regression of plotted points.

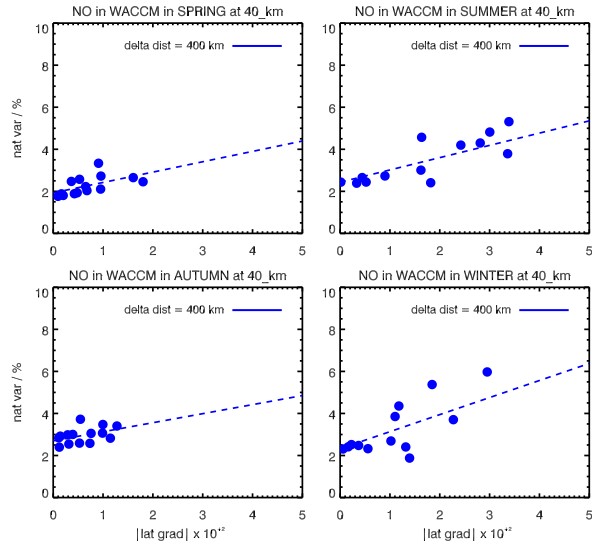

**Figure 7.** Natural variability of NO as function of latitudinal gradients for 400 km distance separation at 40 km altitude in four seasons in WACCM model.





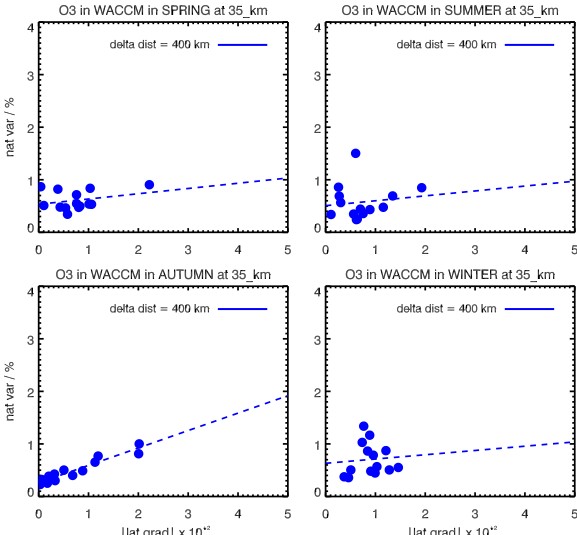

**Figure 8.** Natural variability of $O_3$ as function of latitudinal gradients for 400 km distance separation at 35 km altitude in four seasons in WACCM model.

whole sample, in order to increase the significance of the statistics. Together with regression coefficients values, we provide a software, which takes as input the species name (among 30 available), the distance and time mismatch chosen, the latitude
190 band, the value of latitudinal gradients of his data and the height, and as an output the user obtains a value of the natural variability of the gas, at altitude given, in latitude band given, for mismatch criteria given. If the validation study is perfomed in latitude domain larger then 10 degrees, than the values in correspoding 10-degree bands should be added quadratically.

## 7 Discussion

In the sense of the devil's advocate, we try to raise possible objections against our method and try to rebut them. Since we
195 do not use model data directly but only differences between model data, additive model biases cancel out. Critical minds might plead that there still could be multiplicative biases in the model data, which would affect our statistics of differences. These are, however, not harmful either, if the gradient-related parametrization is used instead of the latitude-month related parametrization. The reason is roughly this. A model bias affects the horizontal gradients in the same way as the differences used for our statistics. Thus, also the effect of a multiplicative model bias cancels out.
200  An obvious objection to our method is that the model data used cover only a short time period and might not be inductively generalizable towards other time periods. We agree that, due to the annual cycle, the typical meteorological regimes are shifted in latitude over the year. But again, when the gradient-related parametrization is used, most likely the statistics of the correct meteorological regime is chosen, even if it is found at different latitudes than the validation experiment. The explanation of this



is that the natural variability of mixing ratios of most trace species is predominantly driven by the latitudinal gradients. It goes
without saying that this does not hold for fast reacting species, and particularly such which are in a photochemical equilibrium.

These parametrizations should not be used where polar vortex dynamics may play a role or for spatial mismatches beyond
1000 km and temporal mismatches beyond 5 hours; but these situations are not the preferred validation scenarios anyway.

An application of the method to ozone and temperature fields will be provided in upcoming validation papers of the Version
8 of MIPAS data.

## 210  8  Conclusions

In validation exercice, a universal excuse used to explain the residual discrepancy between the data is the natural atmospheric
variability due to the imperfect collocations. This work is the first attempt to quantify this atmospheric variability for big
sample of atmospheric constituents and provide the user with a tool to substract from the residual variability the part coming
from natural atmospheric variability. The fields of natural atmospheric variability as function of distance and time mismatch
were calculated out of high-resolved BASCOE model data. The variability data were described by an easy-to-use regression
function and the regression coefficients are provided to the community, together with the software that calculates for given gas,
latitudinal gradient, height, and collocation criteria the value of correspoinding natural variability. An independence of linear
approximation of the natural variability as function of latitudinal gradients from season was demonstrated on WACCM model
data.

*Code and data availability.*  The regression coefficients of the parametrisation on latitudes for autumn season and the software for calculating
the variability as function of latitudinal gradients are stored in https://publikationen.bibliothek.kit.edu/1000137514.

*Data availability.*  Whole Atmosphere Community Climate Model (WACCM) data can be downloadad from ACOM website:
https://www2.acom.ucar.edu/gcm/waccm. WACCM Forecast Maps are available on https://www.acom.ucar.edu/waccm/forecast/.
WACCM at the NCAR/UCAR Research Data Archive can be found at https://rda.ucar.edu/datasets/ds313.6/#!description.

*Author contributions.*  AL did the statistical analysis and developed the parametrization and re-parametrisation. TvC identified the problem,
suggested the solution, supervised the work and contributed to the text. UG ensured the implementation of statistical calculations out of
model fields. QE and SH provided the model fields and guidance to their use, for BASCOE and WACCM models, respectively.

*Competing interests.*  TvC is associate editor of AMT but has not been involved in the evaluation of this paper.



*Acknowledgements.* This work was performed as part ESA VACCUM'R Project. The authors are thankful to Claus Zehner for very effective

supervision of the project. The WACCM data were provided by WACCM forecast team including Simone Tilmes and Douglas Kinnisoni, and website/technical team including Carl Drews and Garth DAttilo. ISSI financed and hosted the 2d Meeting of SPARC TUNER (Toward UNified Error Reporting) Consortium, where the intermediate results of this work were presented and critically accessed by the members of TUNER Consortium. The first author is particularly thankful to Dr Viktoria Sofieva for numerous helpful discussions.





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
