# Peer review of "Satellite data validation: a parametrization of the natural variability of atmospheric mixing ratios"

_Atmospheric Measurement Techniques, 2021_

## Referee Comment (RC1)

**Short resume**

This paper investigates the typical variability of atmospheric trace gas species using model simulations, with the aim to provide a user-friendly tool to be used in validation studies. The motivations and the theoretical framework are well introduced and the critical discussion of the results is well argumented.
This paper fits the scope of AMT, and it is logically written. From my side, I have some comments on specific aspects and a few technical corrections. A whole revision of the paper for typos is needed.

**Specific comments**

1. The theoretical framework in Sect. 3.1 is well introduced and justified. However, the statement about the stationary increments at lines 58-59 is taken as granted; wouldn't it be better to say that the authors assume that the distribution of the differences does not depend on t, based on literature?

2. Symbols and used variable names.
   I find the mathematical symbols used in the paper sometimes confusing and inconsistent. I would suggest to introduce a symbol/name also for the quantities introduced in Eqs. 4,5,8 and for the atmospheric variability, which can then be referred to in the text and in the figures. The amount of trace gas is referred to as X in Sect. 3.1 and as VMR in other equations. In addition, I find the usage of 'x' for the mismatch in Sect.5 not optimal, especially because $\tau$ was already introduced in Sect.3.1, where $t$ was also assumed to indicate time or space.

3. Regarding the variability values shown, for example, in Fig.1, the authors explained that they increase at northern latitudes due to the presence of the polar vortex: wouldn't it be possible to consider points which have a compatible PV values, as usually done in validation studies? In this respect, the assumed parameterization at line 135 doesn't not apply for high latitude case at large distances, right? You could point this out in the description of Fig.2.

4. Was the period chosen for the simulations with the BASCOE model arbitrary?

5. It is not so clear to me until line 173 that you used only BASCOE simulations for Figs.1-4, is it right?

6. Regarding Figs. 6-8, I was wondering if the discrepancies that can be seen for example for $H_2O$ in spring w.r.t. the other seasons, or for O3 in autumn w.r.t. the other plots are just random or reflect variability in the data set or are related to the restricted chosen time period. I would also reduce the span of both x and y axis of Fig. 8 to better see the dots.

7. Have you seen similar linear dependencies as in Figs. 6-8 when changing the 400 km separation? Possibly, a sentence could be added in this respect.

**Technical corrections**

Two frequent incorrent spelling I found are the word 'stationnary' $\rightarrow$ 'stationary', and the construct 'as a function of': can you please check its usage in the text? In addition, in Sect. 3.1 please replace 'gase' with 'gas'.

P1, l4: 'look-up tables on the natural varaibility' → 'look-up tables of the natural variability'

P1, l5: I would say 'distance and time separation'. Also variability values instead of variabilities.

P1, l6: I would write '...and the season-independence of the linear...'

P1, l14: What does this sentence mean? 'Also different impact of prior information on the result has to be considered'. Do you refer to systematic errors due to the a priori value?

P1, l17: state variable measured → measured state variable

P1, l20: of which → in case

P2, l27: developped → developed

P2, l28: vairability → variability

P2, l30: recepee → recipe

P2, l33: frop → from

P2, l36 usually → generally

P2, l37: delete 'of' after study

P2, l47: what does 'they' refer to? Maybe write 'The model simulations contain...'

P2, l49: geopotential heights → geopotential height

P3, l56: not stationary → non-stationary

P3, Last paragraph of Sect. 3.1: please check typos like 'stydying' or 'conlusions'. '...basing on the shape of the statistics obtained...' → '...based on the shape of obtained statistics...'

I would mention also in the introduction that MIPAS has been chosen as example.

P4, l96: it provides → , which provides

P4, l100: present → presents

P4, l106: I would add at the end of the sentence 'for each latitude band and height'

P5, l117: variation → variations

I would put the sentence at lines 118-119 after Eq. 6, right before 'We are not considering...' and add a reference to Sect. 6 where the software is described.

Maybe move the sentence at 139-141 to Sect.3.3.

P6, l159: encounter → tackle.

P7, l166: I would delete 'of the squares', just leave variance of the differences.

P9, l172: delete the repetition of 'were available' and also the spelling of WACCM6 as it was already introduced.

P9, l173 and l175: delete 'The' before Figure.

P9, l173: what do you mean with 'after this manipulation of the data'?

P9, l179: expectalbe → expected.

P9, l186: 'collocation criteria choosen' → 'chosen collocation criteria'

P11, l191: I would say 'at given altitude, latitude band and mismatch'.

P11, l192: check 'corresponding'.

I would move the last sentence at line 208-209 to the conclusions as an outlook.

P12, l211: exercice → exercise.

---

## Author Comment (AC1)

The authors thank the reviewers for the constructive comments which have helped to improve the paper.

**Answer to Reviewer #1.**

*This paper investigates the typical variability of atmospheric trace gas species using model simulations, with the aim to provide a user-friendly tool to be used in validation studies. The motivations and the theoretical framework are well introduced and the critical discussion of the results is well argumented.*
*This paper fits the scope of AMT, and it is logically written. From my side, I have some comments on specific aspects and a few technical corrections. A whole revision of the paper for typos is needed.*
Please find our replies below.

1. *The theoretical framework in Sect. 3.1 is well introduced and justified. However, the statement about the stationary increments at lines 58-59 is taken as granted; wouldn't it be better to say that the authors assume that the distribution of the differences does not depend on t, based on literature?*
   Thank you for the suggestion, the text was changed accordingly.

2. *Symbols and used variable names. I find the mathematical symbols used in the paper sometimes confusing and inconsistent. I would suggest to introduce a symbol/name also for the quantities introduced in Eqs. 4,5,8 and for the atmospheric variability, which can then be referred to in the text and in the figures.*
   We deliberately do not introduce the name for the quantities introduced in the equations 4,5,8, because these are just the intermediate blocks, single elements of the samples, out of which the estimator of natural variability is constructed. While the figures show the values of final estimator, constructed out of these quantities.

   *The amount of trace gas is referred to as X in Sect. 3.1 and as VMR in other equations.*
   This is also done deliberately: in Section 3.1 we are talking about a random variable while VMR refers to its statistical counterpart, VMR being an estimator of X.

   *In addition, I find the usage of 'x' for the mismatch in Sect.5 not optimal, especially because τ was already introduced in Sect.3.1, where t was also assumed to indicate time or space.*
   We presume you are talking about the section 4.2 because there is no $x$ or $\tau$ in the Section 5. Thank you for pointing this out, we have changed $x$ to $\tau$ in the Eq. 7.

3. *Regarding the variability values shown, for example, in Fig.1, the authors explained that they increase at northern latitudes due to the presence of the polar vortex: wouldn't it be possible to consider points which have a compatible PV values, as usually done in validation studies?*
   Some validation studies do not consider compatible PV values in the concern to avoid involve the model data in the validation exercise. In principle, we agree, it could have been done, but the obtained values would drastically depend not only from the season, but also from the year. Our choice was hence to stay with the dependence on the mismatch only.

   *In this respect, the assumed parameterization at line 135 doesn't not apply for high latitude case at large distances, right? You could point this out in the description of Fig.2.*
   This is correct. We have changed the description of the Figure 2 accordingly.

4. *Was the period chosen for the simulations with the BASCOE model arbitrary?*
   Almost arbitrary. There were not so many different data sets available that fulfilled all criteria (resolution, number of gases). We had just to use what was available at that time and looked reasonably representative.

5. *It is not so clear to me until line 173 that you used only BASCOE simulations for Figs.1-4, is it right?*
   Yes, this is correct. We have added in the captures of Figures 1 – 4 that these are calculated out of BASCOE model data.

6. *Regarding Figs. 6-8, I was wondering if the discrepancies that can be seen for example for $H_2O$ in spring w.r.t. the other seasons, or for $O_3$ in autumn w.r.t. the other plots are just random or reflect variability in the data set or are related to the restricted chosen time period.*
   Examination of plots at other heights /separation distance hints toward the random nature of these discrepancies.
   *I would also reduce the span of both x and y axis of Fig. 8 to better see the dots.*
   Done.

7. *Have you seen similar linear dependencies as in Figs. 6-8 when changing the 400 km separation? Possibly, a sentence could be added in this respect.*
   Yes, similar linear dependencies were seen at other heights as well. As suggested, we have added a sentence about it in the text.

*Technical corrections.*
*Two frequent incorrent spelling I found are the word 'stationnary' -> 'stationary', and the construct 'as a function of': can you please check its usage in the text? In addition, in Sect. 3.1 please replace 'gase' with 'gas'.*
Done.

*P1, l14: What does this sentence mean? 'Also different impact of prior information on the result has to be considered'. Do you refer to systematic errors due to the a priori value?*
Yes, this is correct.
All text suggestions were incorporated.

**Answer to Reviewer #2.**

*This paper is acceptable for the publication once the issues noted by the other reviewer have been addressed. I apologize that I was unable to provide a thorough review of the paper. However, the time I was able to spend with the manuscript indicated it to be of sufficient quality and significance and therefore deserving of publication.*

Done.

---

## Author Response (AR2)

The authors thank the reviewer for the constructive comments which have helped to improve the paper. Please find our replies below.

**Answer to Reviewer**

***I still think that a symbol for the natural variability would be better than referring to it as "Rel. nat. var." in the figures, even if you don't want to add a name to Eq. 4 and 5.***

The equations 4 and 5 introduce not the estimator but just the individual member of the sample out of which the estimator is calculated.
As to the symbol for the natural variability, we still think that it would be misleading to introduce a notation that will not be used nowhere except the plots. We would prefer to keep the plots as they are.

All other comments were taken into account and changed accordingly.